# Impact of Point-of-Care Lactate Testing for Sepsis on Bundle Adherence and Clinical Outcomes in the Emergency Department: A Pre–Post Observational Study

**DOI:** 10.3390/jcm13185389

**Published:** 2024-09-12

**Authors:** Sukyo Lee, Juhyun Song, Sungwoo Lee, Su Jin Kim, Kap Su Han, Sijin Lee

**Affiliations:** 1Department of Emergency Medicine, Korea University Ansan Hospital, Ansan-si 15355, Republic of Korea; sukyolee@naver.com; 2Department of Emergency Medicine, Korea University Anam Hospital, Seoul 02841, Republic of Korea; kuedlee@korea.ac.kr (S.L.); icarusksj@gmail.com (S.J.K.); hanks96@hanmail.net (K.S.H.); reonoaz85@gmail.com (S.L.)

**Keywords:** emergency service, lactic acid, sepsis, point-of-care testing

## Abstract

**Background**: The early diagnosis and prompt treatment of sepsis can enhance clinical outcomes. This study aimed to assess the relationship between point-of-care testing (POCT) for lactate levels and both adherence to the Surviving Sepsis Campaign (SSC) guidelines and mortality rates among sepsis patients in the emergency department (ED). We hypothesized that bedside lactate POCT would lead to better clinical outcomes. **Methods**: We conducted a pre–post observational study utilizing data from a prospectively collected sepsis registry. Following the introduction of lactate POCT, lactate levels were determined using both the central laboratory pathway and a POCT device. We then compared the characteristics and clinical outcomes between the periods before and after the introduction of POCT lactate measurement. **Results**: The analysis included a total of 1191 patients. The introduction of bedside lactate POCT led to a significant reduction in the time taken to obtain lactate results (from 53 to 33 min) and an increase in the rate of subsequent lactate measurements (from 82.1% to 88.2%). Lactate POCT did not significantly affect adherence to the overall SSC guidelines bundle (47.5% vs. 45.0%) or reduce 30-day mortality rates (31.1% vs. 31.4%). However, bedside lactate POCT could decrease extremely delayed lactate measurements. **Conclusions**: Bedside lactate POCT successfully reduced the time to obtain lactate results. Although lactate POCT did not lead to improved adherence to the overall SSC guidelines bundle or affect short-term mortality rates in sepsis patients, it may have an advantage in a specific situation such as overcrowded ED where there are subsequent or multiple measurements required.

## 1. Introduction

Sepsis is a life-threatening condition characterized by a dysregulated host response to infection, leading to organ dysfunction [1]. Globally, it represents a significant cause of morbidity and mortality among critically ill patients, causing approximately 11.0 million fatalities in a year [2,3,4]. Early detection of sepsis and the prompt initiation of appropriate treatment are essential for improving clinical outcomes in sepsis patients [5].

Lactate, a byproduct of anaerobic metabolism, is widely recognized as a biomarker for assessing and monitoring the severity of sepsis. Research has demonstrated that lactate levels can predict clinical outcomes in sepsis patients [6,7,8]. The Surviving Sepsis Campaign (SSC) guidelines advise immediate measurement of serum lactate levels in patients suspected of having sepsis, with subsequent measurements if the initial levels are elevated [5]. Traditional central laboratory testing, however, is limited by delays in lactate measurement. Point-of-care testing (POCT) for lactate levels at the bedside has been shown to decrease turnaround times for test results in patients with sepsis [9,10,11,12,13,14]. In the emergency department (ED) setting, various studies have explored the utility of POCT lactate devices for evaluating sepsis severity and the early detection of hyperlactatemia [15,16,17,18,19,20]. Yet, the impact of POCT lactate measurement on clinical outcomes and adherence to SSC guidelines, as compared with central laboratory testing in sepsis patients in the ED, remains to be fully established.

In this study, we aimed to assess the relationship between POCT for lactate levels and both adherence to the SSC guidelines and mortality rates among sepsis patients in the ED. We hypothesized that, compared to central laboratory testing, POCT lactate measurement would enhance adherence to the SSC guidelines and improve clinical outcomes in sepsis patients.

## 2. Materials and Methods

### 2.1. Study Design

This pre–post observational study examined the impact of POCT for lactate measurement on clinical outcomes and adherence to the SSC bundle among sepsis patients in the ED. The study was conducted at the ED of a tertiary teaching hospital with an annual ED attendance of approximately 47,000 patients.

### 2.2. Population

Between January 2016 and September 2022, adults (aged ≥ 18 years) presenting to the ED with a positive quick sepsis-related organ failure assessment (qSOFA) score were screened for inclusion. The qSOFA score comprises three criteria: hypotension (systolic blood pressure ≤ 100 mmHg), tachypnea (≥22 breaths/min), and altered mental state (Glasgow coma scale < 15), with one point assigned for each criterion, leading to a total score ranging from 0 to 3. A positive qSOFA score is defined as being ≥2. The confirmation of inclusion was conducted according to sepsis-3 definition, which is evidence of infection and an increase in the SOFA score by ≥2 points in the ED [1]. The Intelligent Sepsis Management System (i-SMS) at our institution, which is based on qSOFA scores, alerts staff to potential sepsis cases [21]. Upon arrival, triage nurses assess patients’ vital signs and mental status. For those with an initial qSOFA score of ≥2, the digital Order Communication System highlights their name, prompting physicians to measure lactate levels. The i-SMS system also informs the physician of the clinical practice pathway related to sepsis. The clinical practice pathway for sepsis includes a bundle of procedures such as blood culture, lactate measurement, antibiotics administration, and fluid resuscitation according to SSC guidelines. In patients with suspected or confirmed sepsis, physicians can activate the clinical practice pathway which provides the additional order for operational procedure at the time of the subsequent venipuncture. POCT for lactate was utilized for patients enrolled during the period following the study. These patients were prospectively enrolled by an attending ED physician who screened all ED attendees for suspected or confirmed infection and SOFA scores of ≥2. In summary, patients who had (1) positive qSOFA criteria, (2) a suspected or confirmed infection, and (3) an SOFA score of 2 points or higher were included in the present study. During the post-study period, we included patients who or whose legal representatives provided informed consent and those whose initial lactate levels were measured using central laboratory testing in the ED. Patients who died within 2 h of arrival at the ED or had unknown outcomes were excluded. Coronavirus disease-2019 (COVID-19) patients were also excluded because COVID-19 might be a confounder in predicting clinical outcomes in patients with sepsis. We also conducted a subgroup analysis on patients with septic shock. This study adhered to the Declaration of Helsinki and received approval from the Institutional Review Board (IRB) of Korea University Medical Center (IRB no. 2020AS0021). The date of ethical committee approval is 10 December 2019. Written informed consent was obtained from all patients (or their legal representatives) enrolled during the post-period. For those enrolled during the pre-period, informed consent was waived due to the retrospective nature of data collection.

### 2.3. Intervention

Before the introduction of the POCT lactate-measuring device (pre period), lactate levels were exclusively assessed using central laboratory testing. This testing was performed with the Cobas 8000 analyzer (Roche Diagnostics, Mannheim, Germany). On 1 December 2019, the i-STAT CG4+ Cartridge (Abbott, Green Oaks, IL, USA) was introduced as a POCT device for lactate measurement, delivering results within 2 min. Following its introduction, lactate concentrations in sepsis patients were measured using both the central laboratory pathway and the POCT device, with both methods utilizing identical venous samples. The choice of method for subsequent lactate measurements after fluid resuscitation was left to the physician’s discretion without specific instructions. 

Upon receiving informed consent, we collected a whole blood sample from all patients in the post-period who met the inclusion criteria via venipuncture. Immediately after collection, a small drop of blood was applied to a study cassette and analyzed bedside for lactate levels using the POCT device (i-STAT CG4+ Cartridge, Abbott, Green Oaks, IL, USA), with results available within 2 min. For patients with suspected infection and SOFA scores ≥2, the POCT results were promptly shared with the ED physician and a research nurse was assigned to the patient. Further management decisions, including ordering laboratory tests, administering antibiotics, and providing intravenous fluids, were at the discretion of the treating ED physicians. To verify the accuracy of the bedside POCT lactate measurements, results from the POCT device were compared with those from central laboratory testing. In the post-period, clinical decisions were informed by the POCT lactate results.

### 2.4. Outcomes

Trained researchers conducted a structured collection of demographic and clinical variables, which were then recorded in our institution’s sepsis registry. Key time points, such as patient arrival and triage, reporting of central laboratory lactate levels, administration of IV fluids, IV antibiotics, vasopressors, and the time of admission to either a general ward or an ICU were extracted from the electronic health records. These time points are routinely collected and captured by the order communication system. An attending ED physician conducted a structured chart review to determine the final diagnosis and severity of sepsis (sepsis, septic shock), the source of infection, SOFA scores, Acute Physiology and Chronic Health Evaluation II (APACHE II) scores, and in-hospital mortality, all while being blinded to study group and lactate levels. The primary outcome was compliance with the SSC guidelines bundle, including obtaining blood cultures before antibiotic administration, appropriate fluid resuscitation within the first 3 h, antibiotic administration within 3 h, and subsequent lactate measurement if indicated (initial lactate level ≥ 2 mmol/L). Secondary outcomes included the time from ED arrival to antibiotic and vasopressor administration, the volume of IV fluids administered in the first 3 h from ED arrival, and mortality at 7, 14, and 30 days.

### 2.5. Statistical Analysis

Continuous variables were presented as means with 95% confidence intervals (CIs) for parametric data, and medians with interquartile ranges (IQRs) for nonparametric data. They were compared using *t*-tests and Mann–Whitney U tests, as appropriate. The normality of the variables was assessed using the Shapiro–Wilk test. Categorical variables were presented as numbers and percentages, and compared using either the Chi-squared test or Fisher’s exact test, as appropriate. Subgroup patients (septic shock) were examined using the same statistical methods. The agreement between bedside POCT lactate measurements and central laboratory lactate values was analyzed using scatter plots, correlation coefficients, and a Bland–Altman analysis. Statistical analyses were conducted using SPSS (version 25.0; IBM, Armonk, NY, USA) and MedCalc for Windows (version 19.8; MedCalc Software, Mariakerke, Belgium).

## 3. Results

A flowchart of the study population is illustrated in Figure 1. During the retrospective phase of the study, 713 patients were registered in our sepsis registry from January 2016 to November 2019. Of these, 22 patients were excluded due to unknown clinical outcomes (n = 14) or death within 2 h of ED presentation (n = 8). In the prospective phase, from December 2019 to September 2022, 533 patients were registered via the i-SMS and consented to participate in the study. Among these, 33 patients were excluded for reasons of unknown clinical outcomes (n = 10), death within 2 h of ED presentation (n = 7), or diagnosis of COVID-19 (n = 16). Consequently, a total of 1191 patients with sepsis were ultimately included in the study—691 from the pre-period and 500 from the post-period. The median age (IQR) of the patients was 76 (66–82) years, with 58% being male. The all-cause 30-day mortality rate was 31.2%.

The baseline characteristics and clinical outcomes of the study population are detailed in Table 1. Median [IQR] SOFA scores were comparable between the pre-period and post-period groups (8 [6–11] vs. 9 [6–11]; *p* = 0.499), as were the APACHE II scores. Similarly, the baseline lactate levels did not differ significantly between the two groups (2.9 mmol/L vs. 3.1 mmol/L; *p* = 0.073). The incidence of septic shock was 38.8% in the pre-period and 36.6% in the post-period group (*p* = 0.468), and the percentage of male patients was nearly identical (57.9% vs. 57.4%; *p* = 0.905). The median age of patients was higher in the post-period group than in the pre-period group (78 [69–84] vs. 75 [64–81]; *p* < 0.001). The time from arrival to the first lactate result was significantly shorter in the post-period group (33 [21–50] min vs. 53 [42–72] min; *p* < 0.001), (Figure 2A) although the time from arrival to lactate result via the central lab was longer in the post-period group. The median initial serum lactate concentration measured through the central laboratory pathway was similar across both groups (2.9 mmol/L vs. 3.1 mmol/L; *p* = 0.073). Adherence to the overall SSC guidelines bundle was consistent between the pre-period and post-period groups. The rates of appropriate fluid resuscitation within 3 h, obtaining blood cultures before antibiotic administration, and administering antibiotics within 3 h showed no significant differences between the groups. However, the rate of subsequent lactate measurement was significantly higher in the post-period group (82.1% vs. 88.2%, *p* = 0.004). The volume of intravenous fluid administered in the first 3 h was lower in the post-period group (1560 [900–2162] vs. 1360 [860–1899]; *p* < 0.001) (Figure 3A). There was no significant difference in the time from arrival to antibiotics and vasopressor administration between the groups (Figure 4A and Table 1). All-cause mortality at 7, 14, and 30 days showed no significant differences between the pre-period and post-period groups.

The baseline characteristics and clinical outcomes of the patients with septic shock are summarized in Table 2. The study included 268 patients in the pre-period group and 183 patients in the post-period group. Overall, there were no significant differences in underlying diseases between the two groups, with the exception of chronic respiratory disease. The median age was significantly higher in the post-period group (75 vs. 79, *p* = 0.002). The time from arrival to the first lactate result was significantly shorter in the post-period group (50 [39–65] vs. 31 [21–44], *p* < 0.001) (Figure 2B), although the time from arrival to lactate result via the central lab was longer in the post-period group. The median initial serum lactate concentration measured through the central laboratory pathway was comparable between the groups (5.3 mmol/L vs. 6.0 mmol/L; *p* = 0.472). Adherence to the SSC guidelines overall bundle was similar between the pre-period and the post-period groups. The volume of fluid administered in the first 3 h was significantly lower in the post-period group (2045 [1492–2507] mL vs. 1860 [1360–2360] mL; *p* = 0.004) (Figure 3B). The time from arrival to antibiotic and vasopressor administration showed no significant difference between the groups (Figure 4B and Table 2). All-cause mortality at 7, 14, and 30 days showed no significant difference between the pre-period and post-period groups.

The correlation between POCT and central lab lactate levels was 0.99 (*p* < 0.001), as shown in Figure 5. The mean difference between the POCT and central lab lactate readings was 0.5 mmol/L. Bland–Altman plot analysis revealed that the majority of differences fell within a range of −0.9 to 1.9 mmol/L.

## 4. Discussion

In this study, the introduction of bedside POCT for lactate measurements was linked to a significant decrease in the time to initial lactate results and the total volume of IV fluids administered within the first 3 h in patients with sepsis. It also correlated with an increased frequency of subsequent lactate measurements. However, bedside POCT for lactate did not lead to better adherence to the overall SSC guidelines bundle, nor did it reduce the time from arrival to antibiotic or vasopressor administration. Additionally, bedside lactate measurements were not associated with the 7-day, 14-day, or 30-day mortality rates. These findings were consistent among patients with septic shock. 

While lactate measurement is a key component of the SSC guidelines, a 20 min reduction in time to test result did not translate into improved clinical outcomes. A prior study found that the introduction of bedside POCT for lactate was associated with a decrease in in-hospital mortality (from 19% to 5%; *p* = 0.02) and ICU admissions (from 51% to 33%; *p* = 0.02) [22]. The direct impact of earlier lactate measurement on mortality reduction remains uncertain. Notably, the 30-day mortality rate in our study was much higher than the in-hospital mortality reported in the previous study (31.1–31.4% vs. 5–19%). The incidence of septic shock and the SOFA scores in our study were also higher, indicating a greater clinical severity among our study population compared to that in the previous study. This difference in disease severity could partially explain why the bedside POCT lactate measurement did not significantly affect the clinical outcomes. In the earlier study, the time to first lactate results was reduced by 88 min following the POCT device’s introduction (from 122 to 34 min; *p* < 0.001) [22], suggesting that implementation of the POCT lactate measurement was associated with improved survival only when the time to lactate measurement was delayed with the central laboratory pathway. Contrary to the findings of the previous study, our research observed a significant difference in IV fluid administration between the pre- and post-period groups. In our study, bedside POCT lactate measurements were associated with a reduction in the volume of IV fluids given to patients with sepsis and septic shock. As excessive fluid resuscitation can be detrimental in critically ill patients, optimizing IV fluid administration is crucial to minimize the negative impact of fluid overload in patients with sepsis and septic shock [23,24].

According to the Sepsis-3 definitions, adult patients with suspected infections who meet the criteria for a positive qSOFA score can be quickly identified as having a higher likelihood of experiencing poor outcomes typical of sepsis [1]. However, the latest 2021 SSC guidelines advise against using the qSOFA score as the sole method for sepsis screening [4]. Our study utilized a qSOFA-based screening program to identify sepsis patients in line with the Sepsis-3 definitions [21]. Since our inclusion criteria were limited to qSOFA-positive patients, these individuals might have faced poorer clinical outcomes compared to the broader population of sepsis patients not screened by qSOFA criteria alone. Previous research has indicated that a qSOFA-negative result at triage correlates with lower adherence to SSC guidelines [25]. In our study, no difference in mortality and SSC bundle adherence was observed between the pre- and post-period groups, which may be partially attributed to our focus on qSOFA-positive sepsis patients, who are at higher risk of adverse outcomes. Further research is warranted to determine whether POCT lactate measurement affects mortality and SSC bundle adherence among qSOFA-negative sepsis patients.

Because the POCT lactate device can deliver results within minutes, it enables clinicians to obtain immediate feedback when necessary. Therefore, POCT lactate measurements may offer a more practical alternative to central pathway lactate measurements for guiding resuscitation efforts in patients with sepsis. In our study, the rate of subsequent lactate measurements was higher in the group assessed after the introduction of POCT. Previous research has shown that early POCT lactate measurements, due to their shorter turnaround times, could enhance the cost-effectiveness of treatment by reducing instances of overtreatment and unnecessary ICU admissions [26]. Recent studies have explored the effects of restrictive versus standard or liberal fluid management strategies in sepsis patients [27,28]. These studies found that restricting intravenous fluids did not lead to better outcomes compared to standard intravenous fluid therapy. In our study, the volume of IV fluids administered was lower in the post-period group than in pre-period group. From this result, we hypothesize that POCT lactate measurements might influence fluid resuscitation strategies in patients with sepsis and septic shock.

The ANDROMEDA-SHOCK trial showed no difference in 28-day mortality between measuring lactate every 2 h for 8 h and measuring capillary refill time (CFT) every 30 min [29]. The methods used to measure lactate in these studies were either through a central lab or using POCT. Direct comparisons with this study are not possible as the proportion of patients using POCT and the time to report results are unknown. However, in this study, fewer crystalloids were used over 8 h with CFT than with lactate-targeted resuscitation (2359 mL vs. 2767 mL, *p* = 0.01). In our study, the rate of subsequent lactate measurements increased and the amount of crystalloid used decreased after the introduction of POCT lactate. It appears that the more frequently a patient’s condition is assessed by lactate or CFT, the less crystalloid is used. CFT seems to be a reasonable and good alternative to lactate POCT for evaluating the response to resuscitation.

In the pre-period group, there were many outliers that took more than 100 min to report initial lactate results (Figure 2). Overall, the post-study period was a period of chronic overcrowding in the ED, which coincided with the COVID-19 pandemic. Nonetheless, both the time to report initial lactate results and the number of outliers decreased dramatically. The introduction of POCT lactate measurement could assist with rapid decision-making in situations of ED overcrowding such as the COVID-19 pandemic, even though the survival benefit might be unclear.

The post-study period of our study largely coincided with the coronavirus disease-2019 (COVID-19) pandemic in Korea. During this period, patients presenting with fever or respiratory symptoms were screened at triage and placed in a designated quarantine area [30]. Patients with febrile conditions and altered mental status, who could not report respiratory symptoms, were also isolated in separate rooms. As a result, a significant number of patients enrolled in the post-period group were likely treated in isolation. This circumstance may account for the observed delay in obtaining lactate results from the central lab during the period following our study, potentially contributing to delays in completing the SSC guidelines bundle, including measurements of lactate levels. Notably, an increase in in-hospital mortality among patients visiting the ED during the COVID-19 pandemic in Korea has been documented [31,32]. To mitigate the potential impact of the COVID-19 pandemic on clinical outcomes, our study excluded COVID-19 patients with sepsis. However, the quarantine policy and the overall increase in mortality during the period following our study may have influenced our results.

A systematic review encompassing five studies indicated a trend towards reduced in-hospital mortality associated with POCT lactate measurements in sepsis cases [33]. One of these studies reported a significant reduction in hospital stay length, and two others noted a significant decrease in the time to administration of intravenous fluids and antibiotics. The methodologies of these studies varied, including pre-post and observational cohort designs, with no RCTs among the studies reviewed. Given that our findings diverge from those of the previous systematic review, we propose that RCTs are necessary to definitively ascertain the impact of POCT lactate measurements on clinical outcomes in patients with sepsis.

Our study has some limitations. First, our study was conducted at a single institution, limiting its generalizability to other settings. Second, the design of our non-randomized, before-and-after observational study can establish associations between the implementation of bedside POCT lactate measurement and clinical outcomes, but cannot prove causality. Third, the clinicians’ awareness of the ongoing study could have introduced a Hawthorne effect, potentially skewing the results in favor of the post-period group. Despite this potential bias, no mortality reduction was observed in the post-period group. Finally, the period following our study coincided largely with the COVID-19 pandemic in Korea. To reduce the potential impact of the COVID-19 pandemic on clinical outcomes, we excluded patients with COVID-19-associated sepsis.

## 5. Conclusions

The implementation of POCT for lactate measurement in our study led to a reduction in the time to obtain test results and increased the frequency of subsequent lactate measurements among adult sepsis patients. However, it did not correlate with changes in mortality rates or overall adherence to the SSC guidelines bundle in these patients. Because lactate POCT measurement could decrease extremely delayed lactate measurements (outliers), it may have an advantage in a specific situation such as overcrowded EDs where there are subsequent or multiple measurements required. Since our study focused on qSOFA-positive sepsis patients, further prospective studies are necessary to determine whether POCT lactate measurement significantly impacts mortality rates and SSC guidelines adherence among the broader population of sepsis patients.

## Figures and Tables

**Figure 1 jcm-13-05389-f001:**
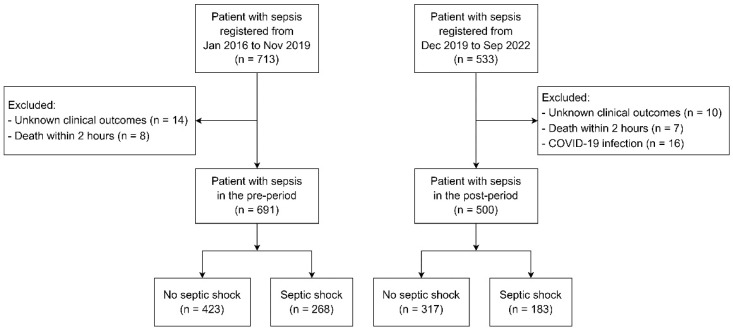
Flowchart of the study population.

**Figure 2 jcm-13-05389-f002:**
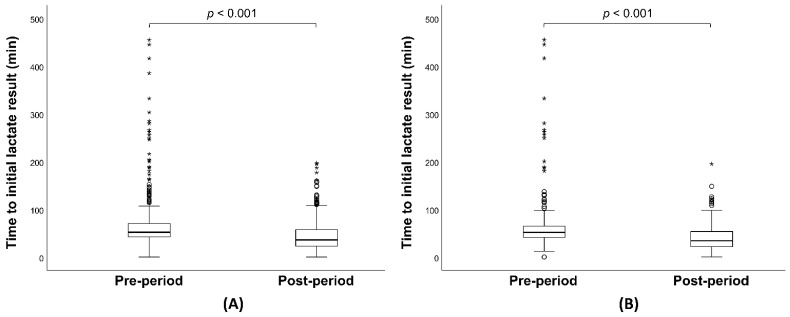
(**A**) Boxplots of time to lactate results administration in overall patients. (**B**) Boxplots of time to lactate results administration in patients with septic shock. The box contains the inter-quartile range and the black line in the middle of the box represents the median. The asterisks mean outliers.

**Figure 3 jcm-13-05389-f003:**
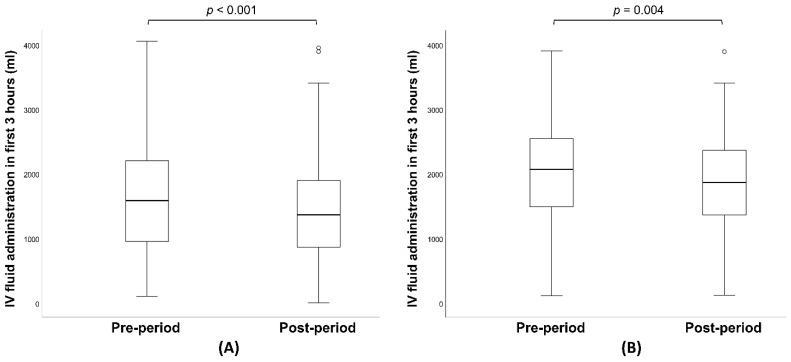
(**A**) Boxplots of time to intravenous (IV) fluid administration within first 3 h in overall patients with sepsis. (**B**) Boxplots of time to IV fluid administration within first 3 h in patients with septic shock. The box contains the inter-quartile range and the black line in the middle of the box represents the median. The tiny circles mean outliers.

**Figure 4 jcm-13-05389-f004:**
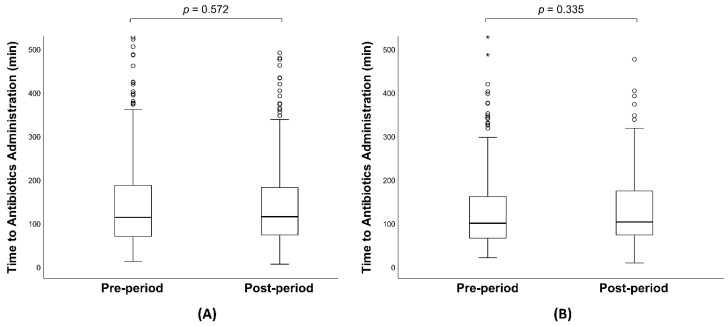
(**A**) Boxplots of time to antibiotics administration in overall patients with sepsis. (**B**) Boxplots of time to antibiotics administration in patients with septic shock. The box contains the inter-quartile range and the black line in the middle of the box represents the median. The tiny circles and asterisks mean outliers.

**Figure 5 jcm-13-05389-f005:**
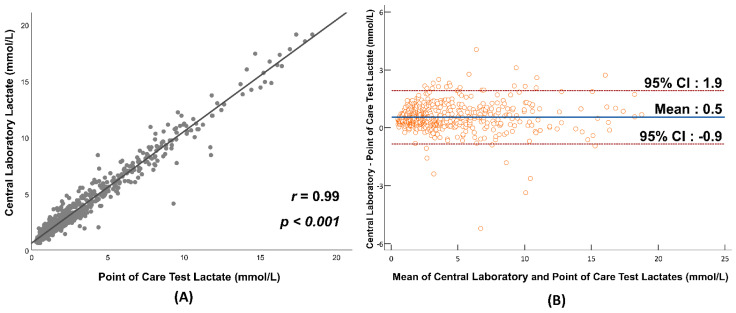
(**A**) Scatterplot of point-of-care test and central laboratory lactates (r = 0.99); (**B**) Bland–Altman plot of central laboratory and point-of-care test lactates. Mean difference between lactates is 0.5.

**Table 1 jcm-13-05389-t001:** Baseline characteristics and clinical outcomes of the study population.

Variables	Pre-Period Group (n = 691)	Post-Period Group (n = 500)	*p* Value
Sex (male), n (%)	400 (57.9%)	287 (57.4%)	0.905
Age, median (IQR)	75 (64, 81)	78 (69, 84)	<0.001
Underlying disease, n (%)			
Diabetes mellitus	269 (38.9%)	212 (42.4%)	0.308
Hypertension	367 (53.1%)	275 (55.0%)	0.679
Chronic liver diseases	38 (5.5%)	27 (5.4%)	0.905
Chronic kidney disease	91 (13.2%)	61 (12.2%)	0.571
Chronic respiratory disease	159 (23.0%)	78 (15.6%)	0.001
Cardiovascular disease	125 (18.3%)	92 (18.4%)	0.962
Malignancy	122 (17.7%)	127 (25.5%)	0.002
Source of infection, n (%)			
Respiratory	456 (66.0%)	316 (63.2%)	0.389
Genitourinary	268 (38.8%)	141 (28.2%)	<0.001
Gastrointestinal	73 (10.6%)	34 (6.8%)	0.026
Other	118 (17.1%)	70 (14.0%)	0.151
Septic shock, n (%)	268 (38.8%)	183 (36.6%)	0.468
Initial serum lactate measured with central lab (mmol/L), median (IQR)	2.9 (1.7, 5.4)	3.1 (2.0, 6.0)	0.073
Time to lactate result (min), median (IQR)			
Time from arrival to first lactate result	53 (42, 72)	33 (21, 50)	<0.001
Time from arrival to lactate result with POCT		33 (21, 50)	
Time from arrival to lactate result with central lab	53 (42, 72)	67 (49, 89)	<0.001
Severity score, median (IQR)			
APACHE II score	18 (14, 22)	19 (14, 24)	0.128
SOFA score	8 (6, 11)	9 (6, 11)	0.499
Surviving sepsis campaign guidelines adherence			
Overall bundle adherence, n (%)	328 (47.5%)	225 (45.0%)	0.410
Appropriate fluid resuscitation in 3 h, n (%)	496 (71.8%)	350 (70.0%)	0.568
Blood culture before antibiotics administration, n (%)	678 (98.1%)	491 (98.2%)	0.735
Antibiotics administration in 3 h, n (%)	543 (78.6%)	386 (77.2%)	0.572
Subsequent lactate measurement (if indicated), n (%)	567 (82.1%)	441 (88.2%)	0.004
Sepsis management			
Time from arrival to antibiotics (min), median (IQR)	111 (69, 183)	114 (72, 178)	0.752
Time from arrival to vasopressor administration (min), median (IQR)	114 (59, 221)	129 (68, 257)	0.051
IV fluid in first 3 h (ml), median (IQR)	1560 (900, 2162)	1360 (860, 1899)	<0.001
All-cause mortality, n (%)			
7-day mortality	136 (19.7%)	97 (19.4%)	0.493
14-day mortality	173 (25.0%)	133 (26.6%)	0.425
30-day mortality	215 (31.1%)	157 (31.4%)	0.950

IQR: interquartile range; POCT: point-of-care testing; APACHE: Acute physiology and chronic health evaluation; SOFA: sepsis-related organ failure assessment; IV: intravenous.

**Table 2 jcm-13-05389-t002:** Baseline characteristics and clinical outcomes of the patients with septic shock.

Variables	Pre-Period Group (n = 268)	Post-Period Group (n = 183)	*p* Value
Sex (male), n (%)	97 (36.2%)	100 (54.6%)	0.051
Age, median (IQR)	75 (61, 82)	79 (69, 84)	0.002
Underlying disease, n (%)			
Diabetes mellitus	117 (44.2%)	75 (41.0%)	0.505
Hypertension	137 (51.7%)	99 (54.1%)	0.617
Chronic liver diseases	23 (8.7%)	10 (5.5%)	0.269
Chronic kidney disease	33 (12.5%)	25 (13.7%)	0.775
Chronic respiratory disease	47 (17.7%)	17 (9.3%)	0.013
Cardiovascular disease	46 (17.4%)	38 (20.8%)	0.390
Malignancy	49 (18.5%)	47 (25.7%)	0.079
Source of infection, n (%)			
Respiratory	164 (61.2%)	106 (57.9%)	0.139
Genitourinary	108 (40.3%)	61 (33.3%)	0.495
Gastrointestinal	42 (15.7%)	19 (10.4%)	0.123
Other	45 (16.8%)	27 (14.8%)	0.602
Initial serum lactate measured with central lab (mmol/L), median (IQR)	5.3 (3.2, 8.4)	6.0 (3.1, 9.4)	0.472
Time to lactate result (min), median (IQR)			
Time from arrival to first lactate result	50 (39, 65)	31 (21, 44)	<0.001
Time from arrival to lactate result with POCT		31 (21, 44)	
Time from arrival to lactate result with central lab	50 (39, 65)	64 (48, 80)	<0.001
Severity score, median (IQR)			
APACHE II score	21 (17, 24)	21 (17, 26)	0.660
SOFA score	11 (9, 13)	12 (10, 13)	0.092
Survival sepsis campaign guidelines adherence			
Overall bundle adherence, n (%)	120 (44.8%)	78 (42.6%)	0.651
Appropriate fluid resuscitation in 3 h, n (%)	181 (67.5%)	118 (64.5%)	0.500
Blood culture before antibiotics administration, n (%)	263 (98.1%)	179 (97.8%)	0.692
Antibiotics administration in 3 h, n (%)	222 (82.8%)	145 (79.2%)	0.335
Subsequent lactate measurement (if indicated), n (%)	199 (76.5%)	150 (82.0%)	0.225
Sepsis management			
Time from arrival to antibiotics (min), median (IQR)	99 (65, 159)	106 (72, 173)	0.619
Time from arrival to vasopressor administration (min), median (IQR)	113 (59, 220)	112 (57, 223)	0.686
IV fluid in first 3 h (ml), median (IQR)	2045 (1492, 2507)	1860 (1360, 2360)	0.004
All-cause mortality, n (%)			
7-day mortality	90 (33.6%)	59 (32.2%)	0.839
14-day mortality	108 (40.3%)	74 (40.4%)	1.000
30-day mortality	132 (49.3%)	85 (46.4%)	0.566

IQR: interquartile range; POCT: point-of-care testing; APACHE: Acute physiology and chronic health evaluation; SOFA: sepsis-related organ failure assessment; IV: intravenous.

## Data Availability

The datasets used and/or analyzed during the current study are available from the corresponding author on reasonable request.

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
