# Peer review of "Impact of Point-of-Care Lactate Testing for Sepsis on Bundle Adherence and Clinical Outcomes in the Emergency Department: A Pre–Post Observational Study"

_jcm, 2024, doi:10.3390/jcm13185389_

Round 1

Reviewer 1 Report

Comments and Suggestions for Authors

Dear Authors,

thank you for your paper. Please find below my questions/suggestions:

1. The abstract is not clear enough - I could not understand what is the difference between the compared groups

2. Actually, who was included? Patients with qSOFA or patients with diagnosed sepsis (increase in SOFA of more than 2 points)? 

3. Please , provide a description of operational procedure for sepsis valid in your hospital. E.g. when sepsis-specific blood samples is collected (in patients positively screened with qSOFA or in patients with SOFA-based diagnosis of sepsis)

4. It looks like in the post-period central lab was less efficient (longer time to lactate result. Did other lab results were also delayed? this should be clarified. 

5. Overall adherence to SSC is rather low while percentages of adherence to fluid resuscitation, blood cultures taking, starting antibiotics  are quite high. What is the reason of that discrepancy ? 

6. You report 7- and 14-day mortality and you have found no difference. I would consider reporting in-ED mortality to seek if any significant difference emerged between pre- i post-period

Comments on the Quality of English Language

Slight improvement in English fluency would be beneficiary. 

Author Response

Comments 1: The abstract is not clear enough - I could not understand what is the difference between the compared groups

Response 1: Thank you for pointing this out. We agree with this comment, thus revised the abstract according to the recommendation. We compared the clinical characteristics, bundle adherence and clinical outcomes between the periods before and after the introduction of POCT lactate measurement.

Comment 2: Actually, who was included? Patients with qSOFA or patients with diagnosed sepsis (increase in SOFA of more than 2 points)? 

Response 2: Thank you for pointing this out. Through our qSOFA-alert system, patients with positive qSOFA criteria were initially screened regardless of the presence of infection. Among them, we included patients who had the evidence of current infection and SOFA score of 2 or higher points. In summary, patients who had 1) positive qSOFA criteria, 2) current infection, and 3) SOFA score of 2 or higher points were included in the present study. We have revised the text to clarify this point.

Comment 3: Please, provide a description of operational procedure for sepsis valid in your hospital. E.g. when sepsis-specific blood samples is collected (in patients positively screened with qSOFA or in patients with SOFA-based diagnosis of sepsis)

Response 3: Thank you for your comment. We have provided the following information about the operational procedure for sepsis in our hospital. The i-SMS system also informs the physicians of the clinical practice pathway related to sepsis. The clinical practice pathway for sepsis includes bundle such as blood culture, lactate measurement, antibiotics administration, and fluid resuscitation according to the SSC guidelines. In patients with suspected or confirmed sepsis, physicians can activate the clinical practice pathway which provides the additional order for operational procedure at the time of the subsequent venipuncture. 

Comment 4: It looks like in the post-period central lab was less efficient (longer time to lactate result. Did other lab results were also delayed? this should be clarified. 

Response 4: Thank you for this very important point. Unfortunately, we do not have data about the time taken to report central laboratory results other than lactate. Therefore, it is difficult to directly compare the time to other central lab results between pre- and post-periods. In our institution, however, blood samples for other central lab results are generally drawn simultaneously with the samples for central lab lactate in most of the ED patients. Although the data about time to other central lab results are not available, we postulate that other lab results were also delayed in the post-period. If the delay exists, that might be partly related with our preventive measures to COVID-19 contagion during the COVID-19 pandemic which coincided with the post-period. Again, we are so sorry for not being able to provide a turnaround time for the other central lab results.

Comment 5: Overall adherence to SSC is rather low while percentages of adherence to fluid resuscitation, blood cultures taking, starting antibiotics are quite high. What is the reason of that discrepancy? 

Response 5: Thank you for pointing this out. Overall adherence to SSC also includes subsequent lactate measurement. The discrepancy was due to the fact that any of the three components was often missing: subsequent lactate measurement, starting antibiotics, and fluid resuscitation. Please see the figures attached below. These figures explain the relatively low percentage of overall bundle adherence to SSC. Again, we really appreciate your helpful comment.

Comment 6: You report 7- and 14-day mortality and you have found no difference. I would consider reporting in-ED mortality to seek if any significant difference emerged between pre- i post-period.

Response 6: Thank you for the helpful comment. Unfortunately, we only have data about 24-hour mortality, not in-ED mortality. Considering that sepsis patients generally stay less than 24 to 36 hours in our ED, we think that 24-hour mortality might substitute for in-ED mortality in the clinical setting of our institution. However, 24-hour mortality was not significantly different between the pre- and post-periods. We are so sorry for not being able to provide in-ED mortality. Again, thank you for pointing this out.

Reviewer 2 Report

Comments and Suggestions for Authors

I read with great interest the manuscript by Lee et. al on the impact of POC lactate testing for sepsis on bundle adherence and outcomes in the ED. The manuscript is sound and well written. However, there are some issues that need to be addressed:

- Line 34-35. In order to give an idea of the global burden of sepsis, authors should provide some more epidemiological details (doi: 10.3390/epidemiologia5030032).

- Line 51. Please replace "we aimed to assess" with "we aimed at assessing".

- Line 89. I believe that this paragraph is not useful, as authors can just report that they defined sepsis according to the sepsis-3 definition. Please remove it.

- Authors should report the date of ethical committee approval.

- In the discussion section, authors should also discuss about the role of capillary refill time as compared to POCT lactate on sepsis and septic shock related outcomes (please see the results of Andromeda Shock Trial).

- Line 259-261. Authors should discuss thoroughly this key point and provide adequate references, as it should be one of the key message of the study. In particular, authors should cite the use of hemodynamic monitoring and the assessment of fluid responsiveness as crucial methods to minimize fluid overload (doi: 10.1186/s13054-022-04255-y - doi: 10.1016/j.tacc.2023.101316).

- Please include the observational non-randomized design as a possible further limitation of the study.

Author Response

Comments 1: Line 34-35. In order to give an idea of the global burden of sepsis, authors should provide some more epidemiological details (doi: 10.3390/epidemiologia5030032).

Response 1: Thank you for your helpful comments. We provided the epidemiological detail for the global burden of sepsis and cited the reference you provided. 

Comments 2: Please replace "we aimed to assess" with "we aimed at assessing".

Response 2: Thank you for the comment. According to your advice, we have revised it.

Comments 3: I believe that this paragraph is not useful, as authors can just report that they defined sepsis according to the sepsis-3 definition. Please remove it.

Response 3: Thank you for the helpful comment. We agree with the point and have removed “2.3.definition” according to your comment.

Comments 4: Authors should report the date of ethical committee approval.

Response 4: Thank you for point this out. Above all, we are so sorry that there was an error in the IRB number. Thus, we have corrected it and have reported the date of ethical committee approval. The exact IRB number for this study is 2020AS0021. Our study was initially approved by our institution on December 10, 2019. After ethical committee approval, the patients in post-period were enrolled from December 11, 2019. As described in the article, written informed consent was obtained from the patients enrolled during post-period. For those enrolled during the pre-period, informed consent was waived due to the retrospective nature of data collection. Thank you again for the important point.

Comments 5: In the discussion section, authors should also discuss about the role of capillary refill time as compared to POCT lactate on sepsis and septic shock related outcomes (please see the results of Andromeda Shock Trial).

Response 5: Thank you for the very informative comments. In our paper, we provided the perspective that subsequent lactate measurement may reduce the use of excessive crystalloid. In this regard, we think it is a very good insight to consider the Andromeda shock trial, which investigated the impact of repeated lactate measurements and capillary refill time assessment on clinical outcomes. We have provided a separate paragraph to address this. Thank you again for the helpful comments.

Comments 6: Authors should discuss thoroughly this key point and provide adequate references, as it should be one of the key messages of the study. In particular, authors should cite the use of hemodynamic monitoring and the assessment of fluid responsiveness as crucial methods to minimize fluid overload (doi: 10.1186/s13054-022-04255-y - doi: 10.1016/j.tacc.2023.101316).

Response 6: Thank you for pointing out the missing reference and suggesting a good article. We have provided and cited related articles according to your comments (reference 23, 24).

Comments 7: Please include the observational non-randomized design as a possible further limitation of the study.

Response 7: Thank you for pointing this out. According to your helpful comments, we provided an additional limitation: “Second, the design of our non-randomized, before-and-after observational study can establish associations between the implementation of bedside POCT lactate measurement and clinical outcomes, but cannot prove causality.”

Round 2

Reviewer 1 Report

Comments and Suggestions for Authors

Dear Authors,

thank you for addressing my comments. I do not have further remarks.